# DnaJC7 specifically regulates tau seeding

Valerie Ann Perez[1], David W Sanders[1], Ayde Mendoza-Oliva[1], Barbara Elena Stopschinski[1], Vishruth Mullapudi[1], Charles L White[2], Lukasz A Joachimiak[1,3], Marc I Diamond[1,4]*

[1]Center for Alzheimer's and Neurodegenerative Diseases, Peter O'Donnell Jr. Brain Institute, University of Texas Southwestern Medical Center, Dallas, United States; [2]Department of Pathology, Peter O'Donnell Jr. Brain Institute, University of Texas Southwestern Medical Center, Dallas, United States; [3]Department of Biochemistry, Peter O'Donnell Jr. Brain Institute, University of Texas Southwestern Medical Center, Dallas, United States; [4]Department of Neurology, Peter O'Donnell Jr. Brain Institute, The University of Texas Southwestern Medical Center, Dallas, United States

**Abstract** Neurodegenerative tauopathies are caused by accumulation of toxic tau protein assemblies. This appears to involve template-based seeding events, whereby tau monomer changes conformation and is recruited to a growing aggregate. Several large families of chaperone proteins, including Hsp70s and J domain proteins (JDPs), cooperate to regulate the folding of intracellular proteins such as tau, but the factors that coordinate this activity are not well known. The JDP DnaJC7 binds tau and reduces its intracellular aggregation. However, it is unknown whether this is specific to DnaJC7 or if other JDPs might be similarly involved. We used proteomics within a cell model to determine that DnaJC7 co-purified with insoluble tau and colocalized with intracellular aggregates. We individually knocked out every possible JDP and tested the effect on intracellular aggregation and seeding. DnaJC7 knockout decreased aggregate clearance and increased intracellular tau seeding. This depended on the ability of the J domain (JD) of DnaJC7 to stimulate Hsp70 ATPase activity, as JD mutations that block this interaction abrogated the protective activity. Disease-associated mutations in the JD and substrate binding site of DnaJC7 also abolished its protective activity. DnaJC7 thus specifically regulates tau aggregation in cooperation with Hsp70.

*For correspondence:
marc.diamond@utsouthwestern.edu

Competing interest: The authors declare that no competing interests exist.

## Editor's evaluation

This important piece of work identified DNAJC7 as a key J-protein that suppresses seeded tau aggregation. The evidence supporting this work is solid. The work will be of interest to cell biologists and neuroscientists.

## Introduction

Neurodegenerative tauopathies are caused by neuronal and glial accumulation of tau protein in amyloid fibrils (*Vaquer-Alicea et al., 2021*). We previously reported that tau has properties of a prion, in which assemblies of defined structure enter a cell, serve as templates for their own replication, and lead to distinct patterns of neuropathology in a process termed 'seeding' (*Frost et al., 2009*; *Kaufman et al., 2016*; *Sanders et al., 2016*; *Sanders et al., 2014*). Recent cryo-electron microscopy studies have revealed distinct fibril morphologies (polymorphs) for several tauopathies, including Alzheimer's disease (AD), progressive supranuclear palsy (PSP), and corticobasal degeneration (CBD) (*Fitzpatrick et al., 2017*; *Shi et al., 2021*; *Zhang et al., 2019*). It has been known for decades that chaperones regulate intracellular protein aggregation. However, the factors that interact specifically with tau to regulate its assembly have not been comprehensively characterized.

Molecular chaperones regulate the folding, maturation, and degradation of proteins (*Hartl et al., 2011*). Heat shock proteins (HSPs) such as Hsp70 and Hsp90 have been reported to non-specifically regulate the folding of tau, alpha-synuclein, and other proteins implicated in neurodegenerative diseases (*Lackie et al., 2017*). The folding and refolding cycle of Hsp70 is regulated by an ATPase cycle, with rapid on/off rates and low substrate affinity in the ATP-bound state and a slow off rate and high substrate affinity in the ADP-bound state (*Mayer and Bukau, 2005*). A separate group of chaperones, the J domain proteins (JDPs), bind myriad substrates and shuttle them to Hsp70 (*Kampinga and Craig, 2010*; *Craig and Marszalek, 2017*). JDPs are defined by a highly conserved ~70 amino acid J domain (JD) that enables interaction with Hsp70. The JD contains a conserved histidine-proline-aspartic acid (HPD) motif required to stimulate Hsp70 ATPase activity and thus enable Hsp70 substrate binding (*Kampinga and Craig, 2010*). JDPs thus play a key role in the Hsp70 protein folding/refolding cycle by conferring substrate specificity to Hsp70.

JDPs have also been reported to directly modulate aggregation of tau and other neurodegenerative amyloid proteins (*Ryder et al., 2022*; *Ayala Mariscal and Kirstein, 2021*). We have previously determined the mode of tau:DnaJC7 interaction (*Hou et al., 2021*). DnaJC7 binds tau with nanomolar affinity and sub-stoichiometrically reduces tau aggregation in vitro. Additionally, DnaJC7 preferentially binds inert tau, a form that does not spontaneously aggregate. We have now tested the specificity of tau binding to DnaJC7 vs. other JDPs and the role of Hsp70 in DnaJC7-mediated regulation of intracellular tau aggregation.

## Results

### Identification of proteins that co-purify with tau aggregates

Tau assemblies of distinct structure ('strains') propagate indefinitely in clonal cultured cells that express the tau repeat domain (RD) containing two disease-associated mutations (P301L/V337M) fused to yellow fluorescent protein (YFP) (*Kaufman et al., 2016*; *Sanders et al., 2014*). To identify factors associated with tau, we studied the insoluble proteome of two clones, termed DS1 (which lacks inclusions) and DS10 (which propagates a unique strain). We first extracted DS1 and DS10 with sarkosyl to identify insoluble material. We boiled the insoluble fraction in SDS and resolved proteins by SDS-PAGE to isolate individual bands for extraction and analysis via mass spectrometry (*Figure 1A*, *Figure 1—figure supplement 1*). We identified 12 unique proteins significantly enriched in DS10 compared to the DS1 control, and 49 proteins found only in the DS10-insoluble fraction (*Figure 1B and C*). These included VCP and Hsp70, which have previously been shown to modulate the tau aggregation process (*Nachman et al., 2020*; *Saha et al., 2023*), and other factors associated with protein quality control, autophagy, and the ubiquitin-proteasome system. As expected, the DS10-insoluble fraction was significantly enriched in tau and YFP. In contrast, the insoluble fraction of DS1 consisted predominantly of RNA-binding proteins and was de-enriched for tau and YFP.

### DnaJC7 knockout reduces clearance and increases inclusion density

We first determined whether any of the top interactors from the proteomic screen would modulate tau aggregate clearance. We utilized a cell line that propagates the DS10 strain with tauRD-YFP expression regulated by a tetracycline-repressible (tet-off) promoter. These cells (henceforth termed OFF1::DS10) constitutively produce large juxtanuclear tau aggregates, whose clearance can be monitored by loss of YFP fluorescent puncta after addition of doxycycline to the cell media to shut off gene expression (*Figure 2A*).

We knocked out a selection of genes enriched in or found only in the DS10-insoluble fraction, plus control genes in the OFF1::DS10 cell lines and monitored tau clearance. Knockout (KO) of certain genes (e.g. VCP, SUMO2, black bars in *Figure 1C*) was lethal, and thus their effects could not be tested. We shut off tau expression for 3 or 5 days to identify cells that had fully cleared aggregates and manually counted cells to quantify the effects of the KO on tau aggregate clearance. OFF1::DS10 cells with DnaJC7 KO had the greatest number of cells still containing tau aggregates after 3 days (~80% containing aggregates) or 5 days (~40% containing aggregates) of repression (*Figure 2B*). This contrasted sharply to the rest of the KO cell lines generated, which cleared most of the aggregates after 5 days of repression. Additionally, confocal microscopy revealed that DnaJC7 KO increased

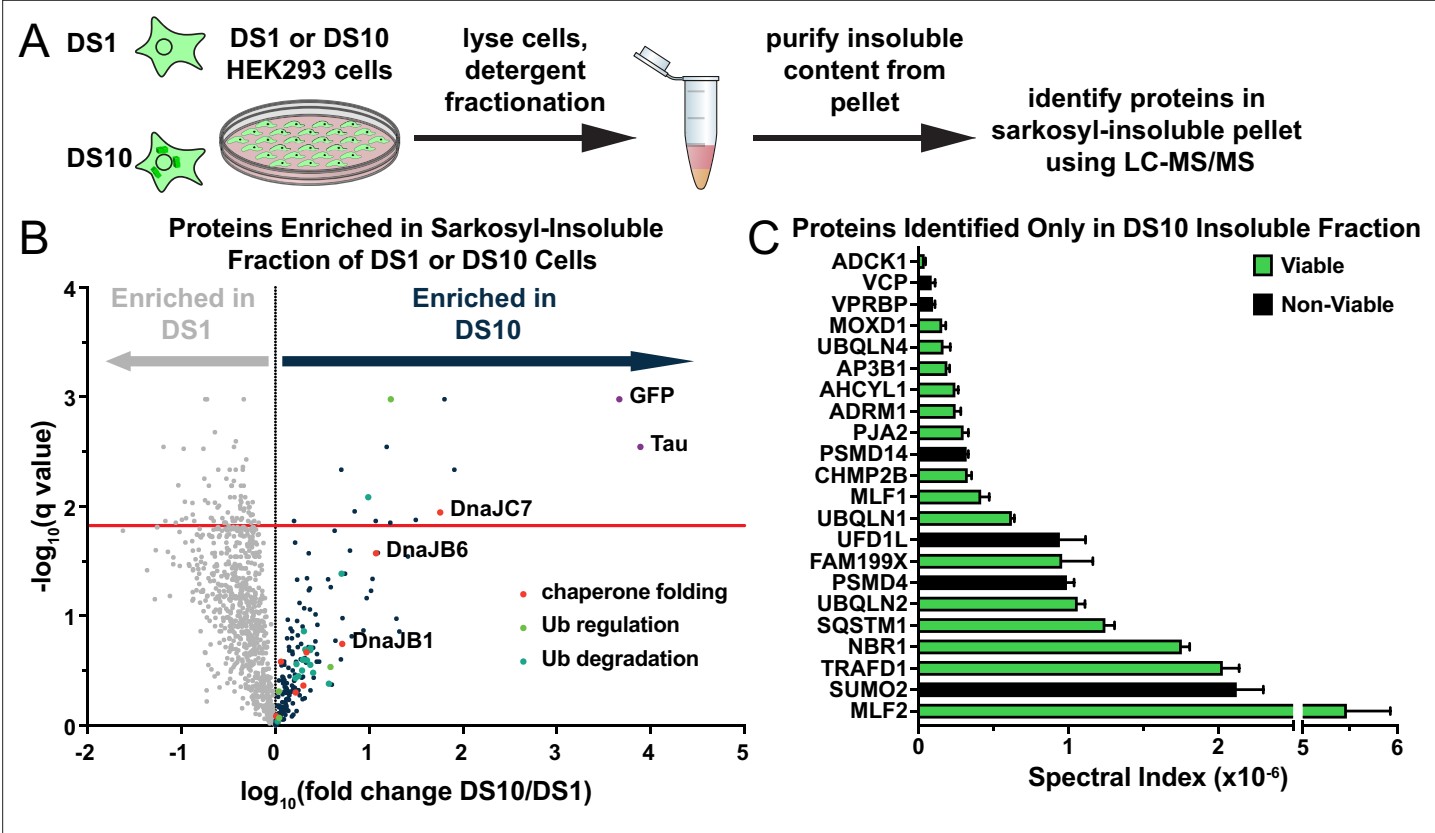

**Figure 1.** A proteomic approach to identify tau aggregate interactors. (**A**) Tau aggregates were partially purified from DS1 and DS10 HEK293 cells expressing tauRD-YFP. Detergent fractionation enabled generation of a sarkosyl-insoluble fraction containing tau aggregates. Proteins were resolved by SDS-PAGE and then extracted from individual lanes for analysis by LC-MS/MS. (**B**) Volcano plot showing proteins enriched in the sarkosyl-insoluble fraction as a fold enrichment from cells expressing tauRD-YFP aggregates (DS10, dark blue dots) over cells expressing tauRD-YFP that does not form aggregates (DS1, gray dots). The red line indicates a false discovery rate of 1.5%. Gene ontology (GO) term enrichment analyses of biological processes is also shown for select GO terms: orange dots, chaperone-mediated protein folding (chaperone folding); green dots, regulation of ubiquitination (Ub regulation); teal dots, ubiquitin-dependent protein catabolic process (Ub degradation). (**C**) Spectral indices for a selection of the proteins identified only in the DS10-insoluble fraction. Viable knockouts are shown as green bars. Non-viable knockouts are shown as black bars. Error bars represent the SEM of three extracted protein SDS-PAGE gel bands.

The online version of this article includes the following source data and figure supplement(s) for figure 1:

**Figure supplement 1.** Partial purification of tau aggregates.

**Figure supplement 1—source data 1.** This source data file contains the original uncropped images for the SDS-PAGE gels shown in *Figure 1—figure supplement 1*.

---

inclusion density and decreased the quantity of non-aggregated (diffuse) tau compared to the nontargeting control (*Figure 2C*). DnaJC7 KO was confirmed by western blot (*Figure 2—figure supplement 1*).

To test for DnaJC7-mediated clearance of invisible seeds, we shut off tau expression for 0–5 days and then restarted expression for 2 days. DnaJC7 KO decreased the rate of tau aggregate clearance, with ~40% of cells contained aggregates after 5 days of repression vs. ~0% of the nontargeting and untreated control cells (*Figure 2D*). The profound effects of DnaJC7 indicated that it likely played a key role in mediating seed clearance.

## DnaJC7 and DnaJB6 uniquely regulate tau aggregation

Given that DnaJC7 KO impeded the clearance of tau aggregates, we next tested whether this would modify seeded tau aggregation. We have previously developed a biosensor cell line that stably expresses tauRD (P301S) linked to the mClover3 or mCerulean3 fluorescent proteins (*Holmes et al., 2014*; *Hitt et al., 2021*). Application of exogenous aggregates induces intracellular tau aggregation

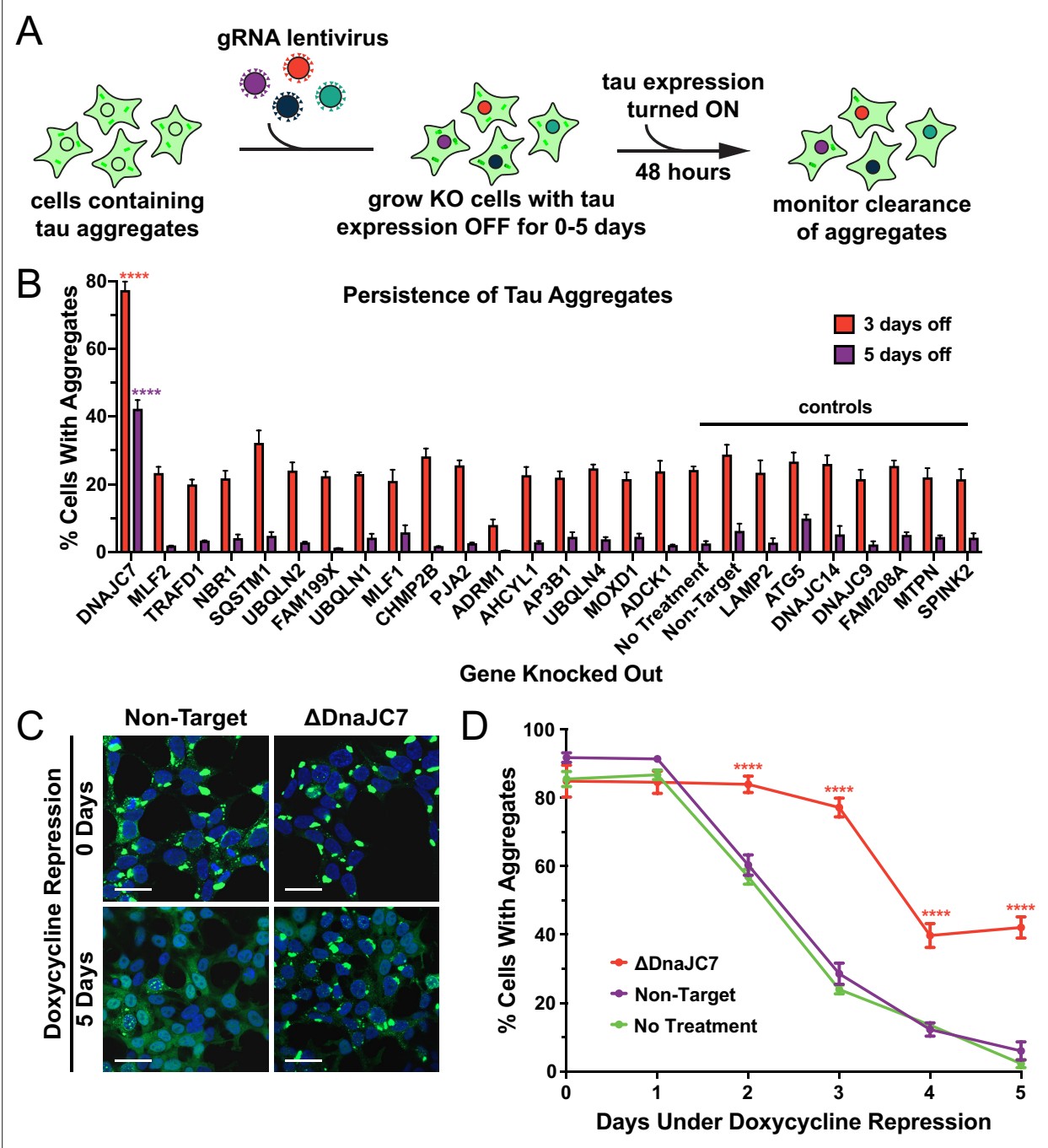

**Figure 2.** DnaJC7 knockout (KO) uniquely extends tau seed lifespan in dividing cells. (**A**) Schematic showing the HEK293 OFF1::DS10 system. A selection of the hits from the proteomics screen was knocked out in these cells. The cells are then allowed to grow with tau expression turned OFF for 0–5days before resuming tau expression. Error bars represent the SEM of six technical replicates. (**B**) The persistence of tau aggregates in OFF1::DS10 cells with the indicated KO was quantified following 3 (orange bars) or 5 (purple bars) days of repressed tau expression. Error bars represent the SEM of six technical replicates. (**C**) Confocal microscopy images showing tau aggregate organization in the DnaJC7 KO and nontargeting control cells following 0 or 5days of repression of tau expression. Scale bars denote 20 μm. (**D**) Extended time course for tau aggregate clearance in the OFF1::DS10 system with DnaJC7 KO (orange) and the nontargeting (purple) and untreated (green) controls. Error bars represent SEM of six technical replicates. *=p < 0.05, **=p < 0.01, ***=p < 0.001, ****=p < 0.0001.

The online version of this article includes the following source data and figure supplement(s) for figure 2:

**Figure supplement 1.** DnaJC7 knockout (KO) in (OFF1::DS10) cells.

**Figure supplement 1—source data 1.** This source data file contains the original uncropped images for the western blot shown in *Figure 2—figure supplement 1*.

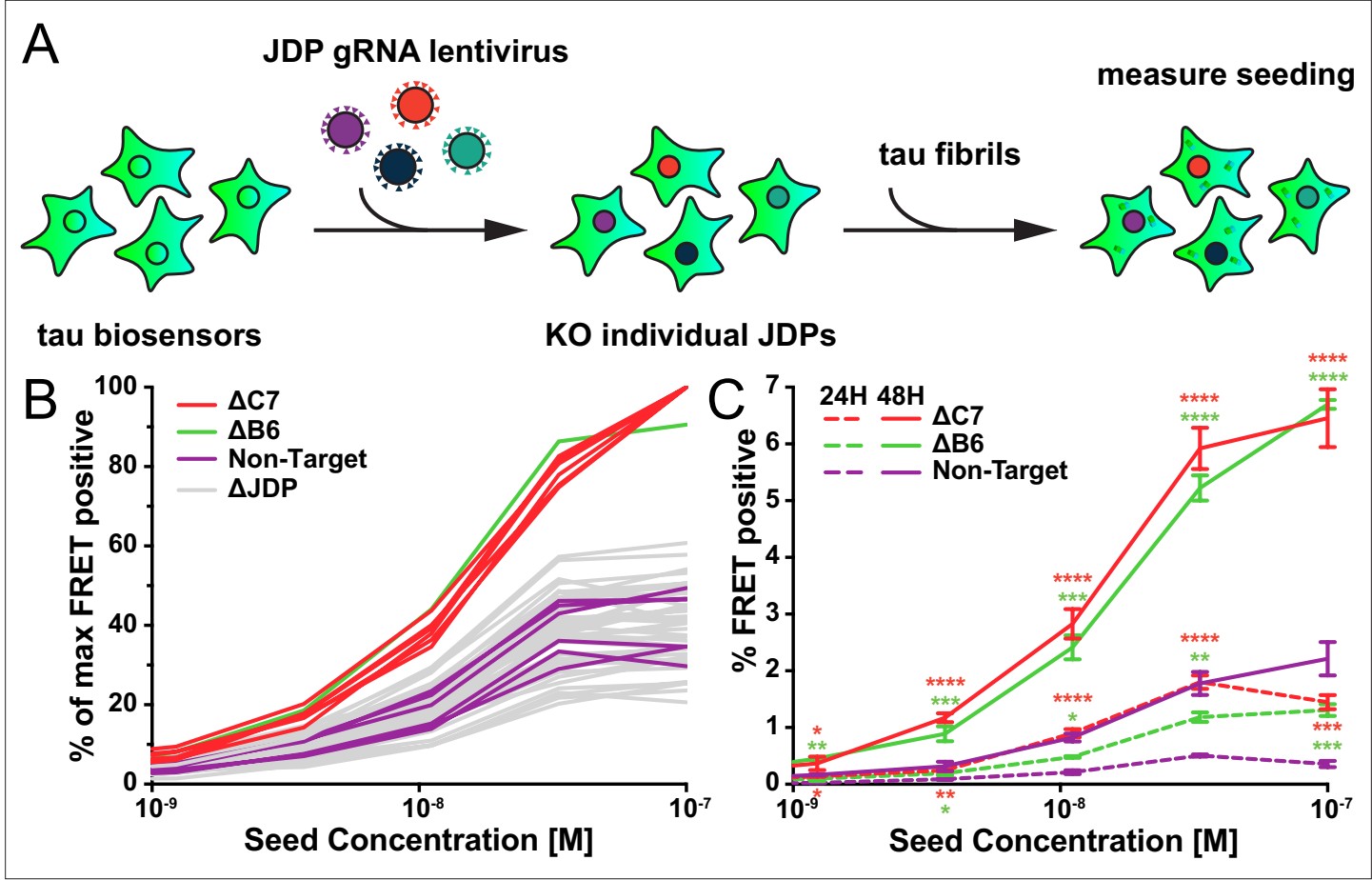

**Figure 3.** A targeted genetic screen for modifiers of transient tau seeding identifies specific J domain proteins (JDPs). (**A**) Schematic showing the HEK293T tau biosensor system consisting of tauRD fused to either mCerulean3 or mClover3 fluorescent proteins. The biosensor cells had each JDP individually knocked out to generate 50 distinct cell lines. Recombinant, sonicated tau fibrils (seeds) were added to the cells to induce seeding of the tauRD constructs, which was detected as a FRET signal via flow cytometry 48hr after treatment. (**B**) Representative data showing the effects of the individual knockouts (KO) of JDPs on tau seeding in biosensor cells, quantified via flow cytometry. Cells were seeded with a dose titration of sonicated tau fibrils. KO of DnaJC7 (ΔC7, orange) and DnaJB6 (ΔB6, green) are highlighted. The remaining JDP KO are denoted as ΔJDP (gray). Each batch of KO cell lines was normalized to the DnaJC7 KO seeding signal and then all batches are plotted together. The seeding assay for the DnaJC7 KO and the nontargeting control (Non-Target, purple) were repeated 10 times. (**C**) Extended time course harvesting of the tau seeding assay for DnaJB6 KO, DnaJC7 KO, and nontargeting control cells harvested at 24hr (24H, dashed lines) and 48hr (48H, solid lines) timepoints. Coloring as in (**B**). Error bars represent SEM of three technical replicates. *=p < 0.05, **=p < 0.01, ***=p < 0.001, ****=p < 0.0001.

The online version of this article includes the following source data and figure supplement(s) for figure 3:

**Figure supplement 1.** All individual groups of the J domain protein (JDP) CRISPR screen.

**Figure supplement 1—source data 1.** This source data file contains the original uncropped images for the western blots shown in *Figure 3—figure supplement 1C*.

that is quantified by fluorescence resonance energy transfer (FRET). Exogenous fibrils bind to heparan sulfate proteoglycans on the cell surface and trigger their own uptake via macropinocytosis. Internalized tau seeds then escape endolysosomal trafficking, enter the cytoplasm, and trigger further intracellular aggregation (*Kolay et al., 2022*). Alternatively, aggregates can be introduced directly into the cytoplasm by cationic lipids such as Lipofectamine 2000, which increases the induced seeding efficiency.

To test the specificity of DnaJC7, we used CRISPR/Cas9 to knock out all known JDPs in the biosensor line individually. We used gRNAs from the Brunello library (*Doench et al., 2016*) in pools of four for each gene to produce 50 unique JDP KO biosensor cell lines (*Figure 3A*). We exposed the KO biosensors to naked seeds at various concentrations. After 48 hr we analyzed seeding in the cells via flow cytometry. Only DnaJC7 and DnaJB6 KO significantly increased tau seeding relative to

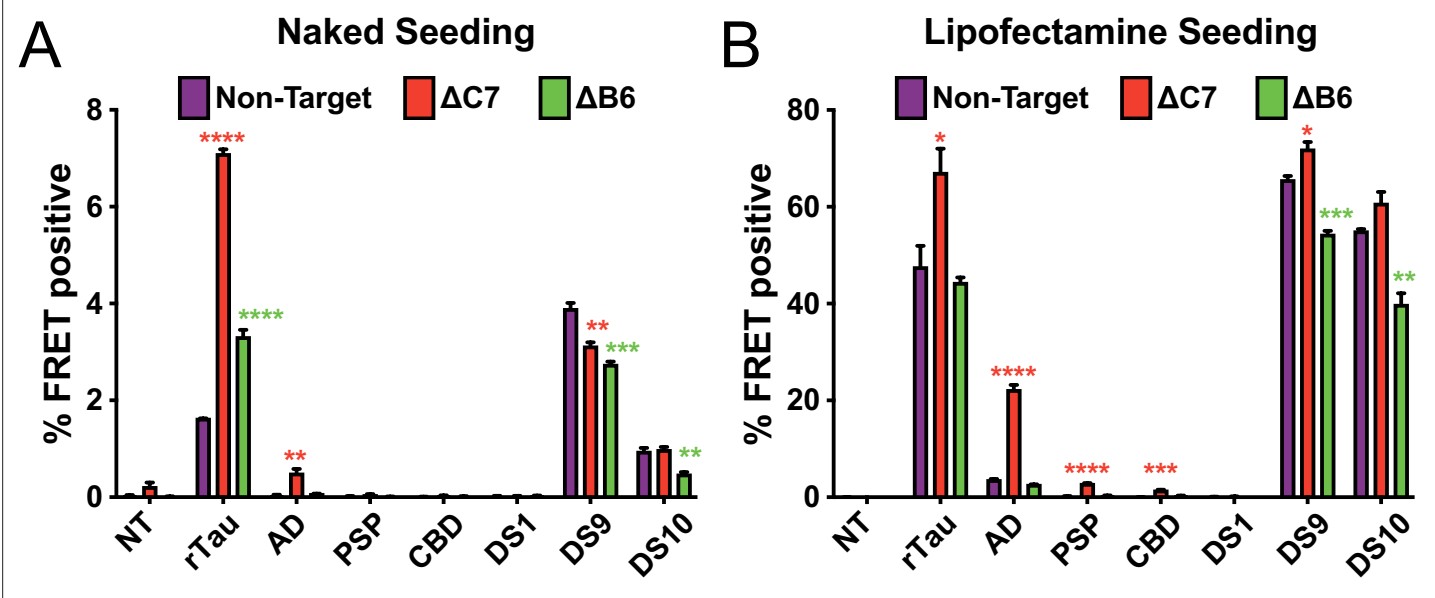

**Figure 4.** DnaJC7 knockout (KO) increases tau seeding across multiple seed sources. Intracellular (**A**) naked or (**B**) lipofectamine-mediated tau seeding in nontargeting control (Non-Target, purple), DnaJC7 KO (ΔC7, orange), and DnaJB6 KO (ΔB6, green) tau biosensor cells. Cells were seeded with sonicated recombinant tau fibrils (rTau); brain lysates from patients with Alzheimer's disease (AD), progressive supranuclear palsy (PSP), or corticobasal degeneration (CBD); and cell lysates of DS1, DS9, or DS10 cells. 100nM or 10nM of recombinant tau were added for naked and Lipofectamine seeding, respectively. 25μL or 5μL of patient brain or lysate or 20μg or 5μg cell lysate were added for naked and Lipofectamine seeding, respectively. NT denotes no treatment with seeds. Error bars represent SEM of three technical replicates. \*=p < 0.05, \*\*=p < 0.01, \*\*\*=p < 0.001, \*\*\*\*=p < 0.0001.

the nontargeting control (**Figure 3B**, full dose titrations shown in **Figure 3—figure supplement 1A**). Western blot analysis later revealed that DnaJC7 was only partially knocked out (**Figure 3—figure supplement 1**). However, even this partial KO of DnaJC7 was sufficient to significantly increase tau seeding.

We also tested whether DnaJB6 and DnaJC7 KO changed the kinetics of intracellular seeding by evaluating seeding at 6, 12, 24, and 48 hr timepoints. We observed no significant seeding in any cell lines at 6 and 12 hr (**Figure 3—figure supplement 1B**). At 24 and 48 hr, DnaJC7 KO enabled more intracellular aggregation than DnaJB6 KO at tau concentrations of 33.3 nM and 11.1 nM. Both KO cell lines exhibited higher seeding than the nontargeting control cell line.

## DnaJC7 KO increases seeding from tauopathy brains

To further characterize DnaJC7 and DnaJB6, we tested their effect on tau seeding with seeds derived from different sources. We treated DnaJC7 KO and DnaJB6 KO cell lines with either recombinant tau fibrils, DS tau strain cell lysates (DS1, -9, or -10) (**Kaufman et al., 2016**), or brain lysates from subjects with Alzheimer's disease (AD), progressive supranuclear palsy (PSP), or corticobasal degeneration (CBD). DS9 propagates a distinct strain of tau aggregates characterized by multiple nuclear inclusions. This contrasts DS10, which features a single large juxtanuclear aggregate. The KO tau biosensor lines were directly treated with 25 μL of brain lysates or transfected with 5 μL of brain lysates using Lipofectamine 2000. Additionally, biosensors were treated with 20 μg total cell lysate from DS1, -9, and -10 or transfected with 5 μg of DS cell lysates using Lipofectamine 2000.

Only DnaJC7 KO significantly increased the seeding of the brain homogenates (**Figure 4A and B**). DnaJC7 KO alone enhanced seeding from naked AD lysate, whereas control and DnaJB6 KO cell lines did not. Thus, the effects of DnaJC7 KO were not constrained by the specific conformation of tau seeds. We also observed that DnaJC7 and DnaJB6 KO induced differential effects on DS9 and DS10 cell lysate seeding in naked vs. Lipofectamine-mediated seeding experiments. When DS9 cell lysate was seeded directly on the cells, DnaJC7 and DnaJB6 KO both resulted in decreased seeding relative to the Non-Target control. By contrast, seeding with the DS10 cell lysate was decreased only in the DnaJB6 KO cell line. Conversely, when DS9 and DS10 cell lysates were transfected into the KO

biosensor lines with Lipofectamine, DnaJC7 KO increased the seeding of DS9 lysate, and DnaJB6 KO decreased seeding for both DS9 and DS10 lysates.

## DnaJC7 overexpression rescues KO tau biosensors

To test for specificity of KO of DnaJC7 in the biosensor cells, we transiently overexpressed DnaJC7 using a Ruby-DnaJC7 fusion with a coding sequence resistant to targeting by the gRNAs used to generate the original DnaJC7 KO cell line (*Figure 5—figure supplement 1A*). The DnaJC7 KO cell line was also remade and full KO was confirmed by western blot (*Figure 5—figure supplement 1B*, vehicle control). Expression of the Ruby-DnaJC7 fusion constructs was confirmed by western blot (*Figure 5—figure supplement 1B*). Ruby-DnaJC7 overexpression in the DnaJC7 KO cells reduced tau seeding relative to Ruby alone and vehicle controls (*Figure 5A*). Overexpression of Ruby-DnaJC7 in the control tau biosensors without DnaJC7 KO modestly reduced tau seeding (*Figure 5B*).

To test the role of DnaJC7 binding to Hsp70, we introduced a point mutation into the Ruby-DnaJC7 fusion that inhibits Hsp70 substrate binding and handoff (*Kampinga et al., 2019*; *Tsai and Douglas, 1996*). The aspartic acid in the HPD motif was mutated to a glutamine (D411Q, henceforth termed HPQ mutant) to preclude DnaJC7's ability to stimulate Hsp70 ATPase activity. Expression of Ruby-DnaJC7 (HPQ) in the DnaJC7 KO biosensors did not rescue the WT seeding phenotype and instead enhanced tau seeding (*Figure 5A*). Surprisingly, overexpression of DnaJC7 (HPQ) in the Non-Targeted control tau biosensors also increased tau seeding, consistent with a dominant negative effect (*Figure 5B*).

## Disease-associated DnaJC7 mutations differentially modulate tau seeding

DnaJC7 mutations cause dominantly inherited amyotrophic lateral sclerosis (ALS) (*Dilliott et al., 2022*). Seventeen mutations have been identified that are distributed across all domains of DnaJC7 (*Figure 5C*). The TPR1 and TPR2A domains have been implicated in binding to EEVD motifs in Hsp70 and Hsp90, respectively (*Assimon et al., 2015*). We have previously found that the TPR2B domain mediates DnaJC7 binding to tau (*Hou et al., 2021*). Additionally, DnaJC7 was recently shown to bind the prion-like domain of the ALS-associated protein TDP-43 and mitigate its ability to phase separate in vitro (*Carrasco et al., 2023*). Although ALS is not traditionally known as a tauopathy, there is evidence of tau pathology in ALS case studies (*Moszczynski et al., 2018*). To test whether these disease-associated mutations can also affect how DnaJC7 modulates tau seeding, we introduced each mutation individually into the Ruby-DnaJC7 construct. We then transiently transfected them into the DnaJC7 KO tau biosensor cell line and quantified changes in tau seeding capacity (*Figure 5—figure supplement 1A*).

Rescue of DnaJC7 KO with most of the mutants suppressed seeding similar to WT. Five mutants increased tau seeding vs. WT (*Figure 5C*). The five inhibitory mutations were in all four domains of DnaJC7, with two located in the JD (R412W and R425K). Neither JD mutant resembled the seeding phenotype of DnaJC7 (HPQ). However, the T341P mutant recapitulated the effect of the DnaJC7 (HPQ) mutant, suggesting a toxic gain of function with effects similar to blocking DnaJC7 stimulation of Hsp70 ATPase activity. We ruled out mutant effects on DnaJC7 expression via western blot (*Figure 5—figure supplement 1C*). The expression of most of the constructs resembled WT, while those that increased tau seeding were slightly reduced in expression, consistent with a gain-of-function effect (*Figure 5—figure supplement 1F*). Thus, DnaJC7 may function within a more complex network of chaperones to control tau aggregation.

To further test the perturbations to the DnaJC7 structure afforded by the ALS-associated mutations, we used Rosetta to calculate the predicted change in the structural stability of the monomer in response to each mutation (*Barlow et al., 2018*). We found no significant correlation between the predicted energy perturbation of the mutations and the resulting seeding phenotype or mutant construct expression (*Figure 5—figure supplement 1E and G*).

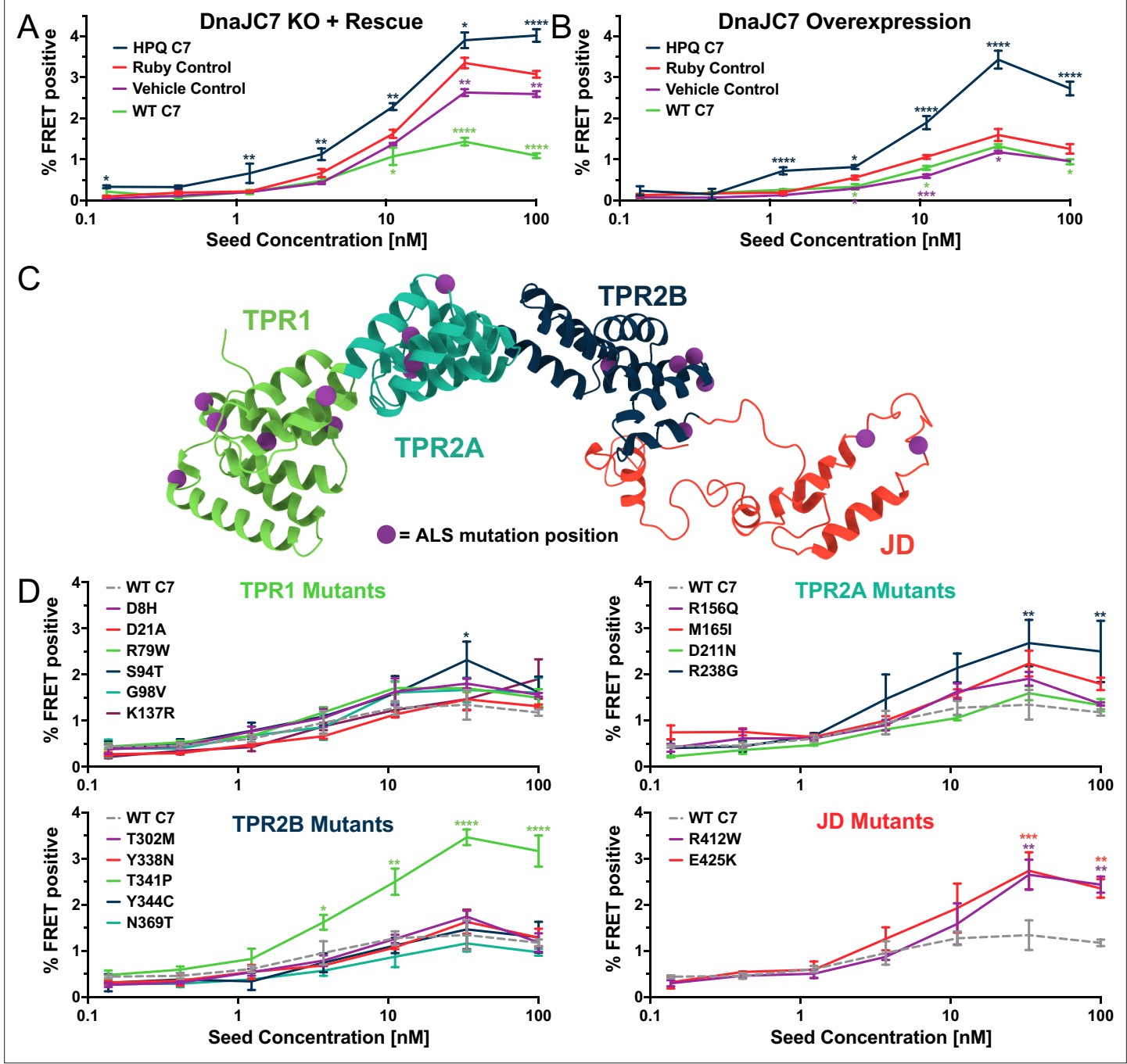

**Figure 5.** DnaJC7 regulates tau seeding in multiple experimental approaches. (**A**) Rescue of DnaJC7 knockout (KO) in tau biosensor cells with either wildtype (WT C7, green) or HPQ mutant (HPQ C7, dark blue) DnaJC7 constructs. The Ruby fluorophore alone (Ruby Control, orange) and a vehicle control (Vehicle Control, purple) were also added to the DnaJC7 KO cells. (**B**) Overexpression of DnaJC7 constructs in control tau biosensor cells. The cells were transfected with the same constructs as in (**A**). (**C**) Model of DnaJC7 with domains colored as follows: TPR1, green; TPR2A, teal; TPR2B, dark blue; JD, orange. Positions of amyotrophic lateral sclerosis (ALS)-associated mutations are shown as purple spheres. (**D**) Rescue of DnaJC7 KO in tau biosensor cells with ALS-associated mutants of DnaJC7 and WT control, sorted by domain location. Rescue with the WT DnaJC7 construct is shown in all domains as a gray, dashed line. Error bars represent SEM of three technical replicates. *=p < 0.05, **=p < 0.01, ***=p < 0.001, ****=p < 0.0001.

The online version of this article includes the following source data and figure supplement(s) for figure 5:

**Figure supplement 1.** Stability and expression of DnaJC7 mutants in tau biosensors.

**Figure supplement 1—source data 1.** This source data file contains the original uncropped images for the western blots shown in *Figure 5—figure supplement 1B and C*.

## Discussion

This study began with an unbiased proteomic screen to identify factors that co-purify with insoluble tau from cells stably propagating a tau strain (DS10). We confirmed the top hits and identified DnaJC7 as a unique regulator of tau aggregation and clearance. The chaperome network is thought to be functionally redundant (*Mok et al., 2018*; *Nachman et al., 2020*; *Yan et al., 2020*; *Gong et al., 2009*). However, when we tested the specificity of all members of the JPD family using a candidate CRISPR/ Cas9 genetic knockdown approach, we identified DnaJC7 and to a lesser degree DnaJB6 as unique regulators of tau aggregation. Finally, we tested newly identified mutations in DnaJC7 which cause ALS and found that those in the JD had dominant negative effects on tau aggregation. This is consistent with the idea that Hsp70-based coordination of DnaJC7 is central to its activity and is linked to protein aggregation in ALS (which usually doesn't feature tauopathy) and to regulation of tau.

### Disease-associated mutations of DnaJC7 differentially affect tau seeding

Pathogenic mutations in several JDPs have previously been linked to multiple heritable neurodegenerative diseases as well as diseases targeting other organ systems (*Zarouchlioti et al., 2018*). These mutations, ranging from missense to inhibitory, to gain-of-function mutants, reduce the ability of the JDPs to bind clients or transfer them to Hsp70, and produce different disease pathologies. Recently, a series of protein truncating variants and missense mutations to the DnaJC7 gene have been identified as causal for ALS (*Dilliott et al., 2022*). The missense mutations were found across all domains of DnaJC7, with most localized to the helix-turn-helix loops on the TPR motifs.

DnaJC7 rescue experiments to test 17 known ALS-associated missense mutations revealed that the T341P mutant recapitulated the seeding profile of the DnaJC7 (HPQ) mutant incapable of stimulating Hsp70 ATPase activity. The T341P mutation is in the TPR2B domain, which we have previously found to be the main site of tau binding on DnaJC7 (*Hou et al., 2021*). Additionally, rescue with the two JD mutants (R412W and R425K) increased seeding relative to the WT DnaJC7 sequence, but both mutants still seeded lower than the T341P and HPQ mutants. Further, the S94T mutant, which may impede Hsp90 binding, also moderately increased tau seeding relative to WT. ALS-associated mutations may inhibit DnaJC7 interaction with other chaperones (e.g. Hsp70 and Hsp90) and thus impair substrate (tau) handoff into the chaperone-mediated protein refolding cycle. This agrees with previous observations that DnaJC7 cooperates with other chaperones to mitigate FUS and TDP-43 toxicity in yeast and in vitro (*Stoltz, 2020*; *Carrasco et al., 2023*).

Additionally, Rosetta modeling found no significant correlation between the predicted energy perturbation of ALS-associated mutations and their resulting seeding phenotype (*Figure 5—figure supplement 1E*). Although the TPR2B mutations Y338N and Y344C were predicted to have the highest destabilizing effects on DnaJC7, they resembled WT in regulating seeding. In contrast, the two JD mutants were predicted to stabilize the DnaJC7 structure but increased seeding. Instead of destabilizing DnaJC7, these mutations exhibited gain-of-function effects likely through dysregulation of interactions within the chaperone network. Disease-associated mutations occur in both DnaJC7 and DnaJB6 (*Dilliott et al., 2022*; *Ruggieri et al., 2015*; *Sarparanta et al., 2020*). This suggests that DnaJB6, DnaJC7, and other JDPs are not functionally redundant.

### DnaJC7 cooperates with Hsp70 to regulate intracellular tau aggregation

Chaperones function in networks to regulate the folding of myriad substrates. The Hsp70 chaperones need JDPs to provide client specificity and activate their ATPase activity. Additionally, co-chaperones such as the TPR containing the Hsc70-Hsp90 organizing protein Hop bridge the activity of Hsp70, which functions on more nascent polypeptide clients, with Hsp90, which folds clients that are closer to their native conformation (*Daniel et al., 2007*). Like Hop, DnaJC7 bridges Hsp70 and Hsp90. However, DnaJC7 also enables retrograde transfer of substrates from Hsp90 to Hsp70, allowing

additional iterations of the Hsp70-Hsp90 cycle (*Brychzy et al., 2003*). DnaJC7 thus uniquely facilitates Hsp70/Hsp90-mediated folding and refolding cycles and could be involved in more complex chaperone networks than the canonical JDPs DnaJB1 or DnaJA2.

DnaJC7 complementation studies in the DnaJC7 KO cell lines indicated that expression of WT DnaJC7 in KO cells rescued the tau aggregation phenotype. In contrast, expression of the DnaJC7 (HPQ) mutant, which abolishes DnaJC7 stimulation of Hsp70 ATPase activity (*Tsai and Douglas, 1996*), increased seeding relative to all controls, indicating an Hsp70-dependent mechanism for DnaJC7 regulation of tau. By preventing substrate handoff to Hsp70, the DnaJC7 (HPQ) mutant proteins may become saturated with tau and unable to shuttle tau between Hsp70 and Hsp90, thus disrupting DnaJC7's role in Hsp70-Hsp90 chaperoning activities and resulting in the apparent dominant negative effects we observed.

## DnaJB6 and DnaJC7 regulation of tau aggregation

JDPs have previously been implicated in various neurodegenerative diseases, with prior work suggesting that DnaJA2, DnaJB1, and DnaJB4 suppress tau aggregation in vitro (*Nachman et al., 2020*; *Irwin et al., 2021*). The literature has also portrayed some JDPs as functionally redundant, as targeted genetic screens knocking out chaperones fail to find many hits in the JDP family that can affect the aggregation of different amyloidogenic proteins (*Mok et al., 2018*; *Kakkar et al., 2016*). We did not identify DnaJA2, DnaJB1, or DnaJB4 as tau aggregation modifiers. This may be because of the model cell line we used.

We also observed a striking difference in the effects of DnaJB6 and DnaJC7 KO in naked vs. Lipofectamine-mediated seeding of DS9 and DS10 cell lysates. Lipofectamine-mediated seeding allows us to bypass the cell's endogenous uptake pathways to transfect seeds into the cell directly. The differential effects of DnaJB6 vs. DnaJC7 KO in the seeding of the DS9 and DS10 cell lysates also hint at different mechanisms for these JDPs' modulation of tau seeding. DnaJB6 KO decreased DS9 and DS10 naked and Lipofectamine-mediated seeding, suggesting that the presence of DnaJB6 in cells may enhance tau seeding. In contrast, DnaJC7 KO decreased DS9 naked seeding and increased DS9 Lipofectamine-mediated seeding. This difference may be attributed to DnaJC7's decreased affinity for tauRD vs. full-length tau, as we have previously observed that DnaJC7 preferentially binds full-length vs. RD tau protein (*Hou et al., 2021*). The different effects of the DnaJB6 vs. DnaJC7 KO in naked vs. Lipofectamine seeding, therefore, hint at potential novel roles of these JDPs in the uptake and processing of certain tau seeds.

DnaJB6 oligomers have been reported to potently suppress polyQ, alpha-synuclein, amyloid beta, and TDP-43 aggregation (*Ruggieri et al., 2015*; *Aprile et al., 2017*; *Gillis et al., 2013*; *Hageman et al., 2010*; *Månsson et al., 2014*). Yet DnaJB6 was not previously known to regulate tau aggregation. It remains unknown how DnaJB6 oligomerization impacts its activity, but experiments on DnaJB6 suppression of polyQ indicated that a serine/threonine-rich region may play a role in the recognition of substrates (*Månsson et al., 2018*). DnaJB6 was not highly abundant in our proteomics screen for tau aggregate interactors, but its KO in tau biosensors increased tau aggregation second only to the DnaJC7 KO. DnaJB6 and its homolog, DnaJB8, have been observed to sub-stoichiometrically inhibit substrate aggregation, suggesting a mechanism based on iterative binding/refolding to aggregation-prone seeds that prevents substrate aggregation (*Gillis et al., 2013*).

We have previously reported that DnaJC7 preferentially bound natively folded tau monomer vs. seed-competent monomer or aggregation-prone mutants (*Hou et al., 2021*). In this study, DnaJC7 KO alone increased tau seeding for all seed sources tested, suggesting an interaction with endogenous tau in a mechanism independent of seed conformation. We hypothesize that DnaJC7 suppresses tau aggregation by binding to inert tau and preventing its templating by exogenous tau seeds. This aligns with our previous finding that DnaJC7 preferentially binds to WT tau, which exists in a more closed conformation, than to the P301L mutant tau, which exists in a more open conformation and

more closely resembles tau seeds. In conclusion, we identified DnaJC7 as a specific regulator of tau aggregation. It binds tau via its TPR2B domain and engages Hsp70 to stabilize the inert monomer.

# Materials and methods

## Key resources table

| Reagent type (species) or resource | Designation | Source or reference | Identifiers | Additional information |
|---|---|---|---|---|
| Cell line (human) | HEK293 | ATCC | CRL-1573 | |
| Cell line (human) | HEK293T | ATCC | CRL-3216 | |
| Cell line (human) | HEK293T tauRD P301S v2L FRET Biosensor | Produced by Diamond Lab | | Stably expresses mClover3 and mCerulean3 tagged tauRD monomers. |
| Cell line (human) | HEK293 DS1 tauRD(P301L-V337M)-YFP | Produced by Diamond Lab | | Stably propagates diffuse tauRD monomers. |
| Cell line (human) | HEK293 DS10 tauRD(P301L-V337M)-YFP | Produced by Diamond Lab | | Stably propagates a unique tau strain. |
| Cell line (human) | HEK293 OFF1::DS10 tauRD(P301L-V337M)-YFP | Produced by Diamond Lab | | Stably propagates a unique tau strain under a tetracycline-repressible promoter. |
| Biological sample (human) | Alzheimer's Disease subject brain | UT Southwestern Alzheimer's Disease Center | | |
| Biological sample (human) | Progressive Supranuclear Palsy subject brain | UT Southwestern Alzheimer's Disease Center | | |
| Biological sample (human) | Corticobasal Degeneration subject brain | UT Southwestern Alzheimer's Disease Center | | |
| Antibody | Rabbit polyclonal anti-DnaJC7 | Proteintech | 11090-1-AP | 1:2000 dilution |
| Antibody | Rabbit polyclonal anti-DnaJB6 | Proteintech | 11707-1-AP | 1:2000 dilution |
| Antibody | Rabbit polyclonal anti-Beta-Tubulin | Proteintech | 10094-1-AP | 1:5000 dilution |
| Antibody | Donkey-anti-rabbit HRP-linked F(ab')2 | Cytiva | NA9340-1ML | 1:5000 dilution (DnaJC7, Beta-Tubulin), 1:4000 dilution (DnaJB6) |
| Antibody | Mouse monoclonal anti-GAPDH | Proteintech | 60004-1-Ig | 1:10000 |
| Antibody | Mouse monoclonal anti-Beta-Actin | Proteintech | 66009-1-Ig | 1:5000 dilution |
| Antibody | Goat-anti-mouse H&L (HRP) | Abcam | ab6789 | 1:10000 dilution (GAPDH), 1:5000 dilution (Beta-Actin) |
| Peptide, recombinant protein | Human tau 2N4R (Full Length WT-tau) fibrils | Produced by Diamond Lab | | MAEPRQEFEVMEDHAGTYGLGDRKDQGGYTMHQDQEGDTDAGLKESPLQ TPTEDGSEEPGSETSDAKSTPTAEDVTAPLVDEGAPGKQAAAQPHTEIPEGTT AEEAGIGDTPSLEDEAAGHVTQARMVSKSKDGTGSDDKKAKGADGKTKIATP RGAAPPGQKGQANATRIPAKTPPAPKTPPSSGEPPKSGDRSGYSSPGSPGTP GSRSRTPSLPTPPTREPKKVAVVRTPPKSPSSAKSRL |
| Recombinant DNA reagent | gRNA-resistant WT DnaJC7 cDNA sequence | gBlock from IDT, cloned by VAP | | The sequence can be found in Materials and methods: Design of a DnaJC7 construct resistant to targeting by used gRNA sequences |

## Identification of aggregate-associated proteins by mass spectrometry

DS1 and DS10 cells were grown to confluency in two T300s per condition. Cells were harvested, pelleted, and washed prior to storage as 4×0.5 T300 pellets at –80°C. For each condition, three pellets were thawed on ice, and each was lysed by trituration in 1 mL ice-cold PBS with 0.25% Triton-X and containing cOmplete mini EDTA-free protease inhibitor cocktail tablet (Roche) at a concentration of 10% wt/vol followed by a 15 min incubation on ice. Aggregatess and nuclei were collected by centrifuging at 1000×$g$ for 15 min followed by resuspension in 400 µL lysis buffer. An Omni-Ruptor 250 probe sonicator was then used at 30% power for thirty 3 s pulses to partially dissolve the pellets. Samples were centrifuged at 250×$g$ for 5 min and the supernatant was set aside as Fraction B. Pellets

were re-homogenized in an additional 400 µL lysis buffer and sonication and centrifugation was repeated. The final supernatant was added to the previous Fraction B (800 µL volume total). A Bradford assay (Bio-Rad) with BSA standard curve was performed and the protein concentrations were calculated for the nine fractions. Protein concentrations were normalized to 1.1 µg/µL. 72 µL of 10% sarkosyl was added to 650 µL of each sample in ultra-centrifuge tubes (Beckman Coulter) and samples were rotated end-over-end at room temperature for 1 hr. Samples were then spun at 186,000×$g$ for 60 min, supernatant was set aside, and pellets were washed with 1 mL lysis buffer prior to an additional 30 min 186,000×$g$ spin. Final pellets were resuspended in 30 µL PBS containing 2% SDS and 2% BME by boiling and trituration. 5 µL of Fraction B supernatants and pellets were loaded onto NuPAGE 10% Bis-Tris gels (Life Technologies) and were run at 150 V for 60 min. Gels were washed 1× with water and were then stained with SimplyBlue SafeStain (Life Technologies). Images of gels were captured using a digital Syngene imager.

For LC-MS/MS-based detection of proteins, 20 µL resuspended Fraction B pellets were run 1 cm onto an Any kD Mini-Protean TGX gel (Bio-Rad) followed by Coomassie Blue staining. Whole lanes were excised using ethanol-washed razor blades and gel samples were cut into 1 mm chunks. Gel pieces were reduced with DTT and alkylated with iodoacetamides (Sigma-Aldrich) and were then digested overnight with trypsin (Promega). Next, excised proteins were subjected to solid-phase extraction cleanup with Oasis HLB plates (Waters). The processed samples were then analyzed by LC-MS/MS using a Q Exactive mass spectrometer (Thermo Electron) coupled to an Ultimate 3000 RSLC-Nano liquid chromatography system (Dionex). Samples were injected onto a 180 µm i.d., 15 cm long column packed with reverse-phase material ReproSil-Pur C18-AQ, 1.9 µm resin (Dr. Maisch GmbH, Ammerbuch-Entringen, Germany). Peptides were eluted with a gradient from 1% to 28% buffer B (80% (vol/vol) ACN, 10% (vol/vol) trifluoroethanol, and 0.08% formic acid in water) over 60 min. The mass spectrometer could acquire up to 20 MS/MS spectra for each full spectrum obtained. Raw mass spectrometry data files were converted to a peak list format and analyzed using the central proteomics facilities pipeline (CPFP), version 2.0.3 (*Trudgian and Mirzaei, 2012*; *Trudgian et al., 2010*). Peptide identification was performed using the X!Tandem and open MS search algorithm (OMSSA) search engines against the human protein database from Uniprot, with common contaminants and reversed decoy sequences appended (*Geer et al., 2004*; *Elias and Gygi, 2007*). Fragment and precursor tolerances of 20 ppm and 0.1 Da were specified, and three miscleavages were allowed. Carbamidomethylation of Cys was set as a fixed modification and oxidation of Met was set as a variable modification.

Label-free quantitation of proteins across samples was performed using SINQ normalized spectral index software (*Trudgian et al., 2011*). Finally, spectral counts were added across triplicates. Proteins with a spectral count greater than 5 in DS10, but not identified in DS1 were reported. To calculate enrichment of proteins in the DS10 samples vs DS1 samples, the spectral counts were first negative $\log_{10}$ transformed. False discovery rate (FDR) analysis was performed using the multiple unpaired t-tests analysis in Prism (GraphPad). The original FDR method of Benjamini and Hochberg was applied, with the desired FDR set to 1.5%. Differences are reported as DS10 spectral index – DS1 spectral index. The -$\log_{10}$(q-value) is reported. Gene ontology analysis was conducted via the Metascape gene annotation and analysis resource (*Zhou et al., 2019*).

## Tau aggregate degradation time courses

OFF1::DS10 cells were treated with two rounds of indicated pooled gRNA lentivirus and were maintained in 24-well plates to assess viability. Two weeks later, a time course examining the decay of tau aggregate seeding activity in the various non-lethal KO was performed as follows: Confluent 24-wells were resuspended in 1 mL media and 3.5 µL cells were re-plated into 200 µL total volume in 96-well plates. Tau RD-YFP expression was turned off using 30 ng/mL doxycycline for 1 day, 2 days, 3 days, 4 days, or 5 days. After 5 days, cells reached confluency and were passaged onto coverslips. 48 hr later, at which point tauRD-YFP expression reached its maximum, cells were fixed and cells containing or lacking inclusions were manually counted. Six replicates of 150+ cells were counted per condition, and averages were calculated. One-way analysis of variance with Bonferroni's correction for multiple comparisons was used to assess statistical significance relative to Non-Target controls.

## CRISPR/Cas9 KO of JDPs in tau biosensor cells

Four human gRNA sequences per gene were selected from the Brunello library (*Doench et al., 2016*). A single nontargeting human gRNA sequence was used as a negative control. For all gRNA sequences not beginning with guanine, a single guanine nucleotide was added at the 5′-end of the sequence to enhance U6 promoter activity. DNA oligonucleotides were synthesized by IDT DNA and cloned into the lentiCRISPRv2 vector (*Sanjana et al., 2014*) for lentivirus production. The plasmids for the four gRNAs for each gene were pooled together to a final concentration of 40 ng/µL and used to generate lentivirus.

Lentivirus was produced as described previously (*Stopschinski et al., 2018*). Briefly, HEK293T cells were plated at a concentration of 100,000 cells/well in a 24-well plate. 24 hr later, cells were transiently co-transfected with PSP helper plasmid (300 ng), VSV-G (100 ng), and gRNA plasmids (100 ng) using 1.875 µL of TransIT-293 (Mirus) transfection reagent in OptiMEM to a final volume of ~30 µL. 48 hr later, the conditioned medium was harvested and centrifuged at 1000 rpm for 5 min to remove dead cells and debris. For transduction, 30 µL of the virus suspension was added to HEK293T tau biosensor cells at a cell confluency of 60% in a 96-well plate. 48 hr post-transduction, infected cells were treated with 1 µg/mL puromycin (Life Technologies, Inc) and maintained under puromycin selection for at least 10 days after the first lentiviral transduction to ensure KO of the individual JDPs before conducting experiments.

## Recombinant tau seeding assay in JDP KO tau biosensor cells

The individual JDP KO cell lines were tested in batches of 10 KO cell lines, including the DnaJC7 and nontargeting control cell lines for each batch. Each batch was assayed in biological duplicate. For all experiments, cells were plated in 96-well plates at 20,000 cells/well in 100 µL of media. 24 hr later, the cells were treated with 50 µL of a heparin-induced recombinant tau fibril dilution series in technical triplicates. Prior to cell treatment, the recombinant tau fibrils were sonicated for 30 s at an amplitude of 65 on a Q700 Sonicator (QSonica). A threefold dilution series of the sonicated fibril concentrations ranging from 100 nM to ~15.2 pM and a media negative control was added to the cell media. 48 hr after treatment with tau, the cells were harvested by 0.05% trypsin digestion and then fixed in PBS with 2% paraformaldehyde (PFA) for 10 min. The cells were then washed and resuspended in 150 µL of PBS.

## Flow cytometry of tau biosensor cells

A BD LSRFortessa was used to perform FRET flow cytometry. To measure mCerulean3 and FRET signal, cells were excited with the 405 nm laser and fluorescence was captured with a 405/50 nm and 525/50 nm filter, respectively. To measure mClover3 signal, cells were excited with a 488 laser and fluorescence was captured with a 525/50 nm filter. To quantify FRET, we used a gating strategy where mCerulean3 bleed through into the mClover3 and FRET channels was compensated using FlowJo analysis software, as described previously (*Furman et al., 2015*). FRET signal is defined as the percentage of FRET-positive cells in all analyses. For each experiment, 10,000 mClover3/mCerulean3 double-positive cells per replicate were analyzed and each condition was analyzed in triplicate. Data analysis was performed using FlowJo v10 software (Treestar). One-way analysis of variance with Dunnett's correction for multiple comparisons was used to assess statistical significance relative to Non-Target controls in GraphPad Prism.

## Preparation of brain and DS cell lysates

Frontal lobe sections of 0.5 g from AD, PSP, and CBD patients were gently homogenized at 4°C in 5 mL of TBS buffer containing cOmplete mini EDTA-free protease inhibitor cocktail tablet (Roche) at a concentration of 20% wt/vol using a Dounce homogenizer. Samples were centrifuged at 21,000×$g$ for 15 min at 4°C to remove cellular debris. Supernatant was partitioned into aliquots, snap-frozen in liquid nitrogen, and stored at –80°C.

DS1, DS9, and DS10 cell pellets were lysed by resuspension in cold 0.05% Triton in PBS containing cOmplete mini EDTA-free protease inhibitor tablet (Roche) at a concentration of 10% wt/vol followed by incubation in the lysis buffer for 20 min on ice. Homogenates were then clarified by centrifugation at 4°C at a speed of 500 RCF for 5 min followed by 1000 RCF for 5 min. The supernatant was then

isolated, and total cell lysate protein concentrations were measured using the Pierce BCA Protein Assay Kit (Thermo Fisher).

## Biosensor seeding with brain and DS cell lysates

Seeding with the brain and DS cell lysates was conducted on the DnaJC7, DnaJB6, and nontargeting KO tau biosensor cell lines. For all Lipofectamine seeding experiments, cells were plated in 96-well plates at 20,000 cells/well in 130 µL of media. 24 hr later, the cells were treated with 20 µL of Lipofectamine complexes. The complexes are generated by preparing a Lipofectamine in OptiMEM (Gibco) master mix consisting of 1 µL Lipofectamine 2000 (Gibco) and 9 µL OptiMEM per well. The seeding material is prepared in a separate master mix for each lysate tested and consists of 5 µL of brain lysate and 5 µL of OptiMEM per well or 5 µg total protein of DS cell lysate and OptiMEM to 10 µL. The Lipofectamine and lysate master mixes are combined in a 1:1 ratio and incubated for 30 min at room temperature. The final master mix is then distributed to triplicate wells for the three KO cell lines, with each well receiving 20 µL of treatment. A Lipofectamine control and OptiMEM-only negative control were also generated.

For all naked seeding experiments, cells were plated in 96-well plates at 10,000 cells/well in 100 µL of media. 24 hr later, the cells were treated with 50 µL of seeding complexes. The seeding material is prepared in a separate master mix for each lysate tested and consists of 25 µL of brain lysate and 25 µL of complete media per well or 20 µg total protein of DS cell lysate and media to 50 µL per well. The final master mix is then distributed to triplicate wells for the three KO cell lines, with each well receiving 50 µL of treatment. A media-only negative control was also generated.

48 hr after treatment, the cells were harvested by 0.05% trypsin digestion and then fixed in PBS with 2% PFA for flow cytometry. One-way analysis of variance with Dunnett's correction for multiple comparisons was used to assess statistical significance relative to Non-Target controls in GraphPad Prism.

## Design of a DnaJC7 construct resistant to targeting by used gRNA sequences

N-terminal Ruby fusion constructs of DnaJC7 were designed to be resistant to targeting by the four gRNAs used to generate the DnaJC7 KO biosensor line. Alternative codon sequences were used at the four gRNA sites to generate a distinct cDNA sequence that maintained the same amino acid sequence. The cDNA sequence is copied below, with alternative codon sequences underlined:

atggcggctgccgcggagtgcgatgtggtaatggcggcgaccgagccggagctgctcgacgaccaagaggcgaagaggga
agcagagactttcaaggaacaaggaaatgcatactatgccaagaaagattacaatgaagcttataattattatacaaaagccatagatatgt
gtcctaaaaatgctagctattatggtaatcgagcagccacgctgatgatgctgggccgcttccgggaagctcttggagatgcacaac
agtcagtgaggttggatgacagttttgtccggggacatctacgagagggtaaatgccatctgagcctcgggaatgccatggcagcatg
tcgcagcttccagagagccctagaactggatcataagaacgcgcaggcgcagcaggaattcaagaatgctaatgcagtcatggaat
atgagaaaatagcagaaacagattttgagaagcgagattttcggaaggttgtttctgcatggaccgtgccctagaatttgcccctgc
ctgccatcgcttcaaaatcctcaaggcagaatgtttagcaatgctgggtcgttatccagaagcacagtctgtggctagtgacattctacgaa
tggattccaccaatgcagatgctctgtatgtacgaggtctttgcctttattacgaagattgtattgagaaggcagttcagtttttcgtacag
gctctcaggatggctcctgaccacgagaaggcctgcattgcctgcagaaatgccaaagcactcaaagcaaagaaagaagatgggaata
aagcatttaaggaaggaaattacaaactagcatatgaactgtacacagaagccctggggatagaccccaacaatataaaaacaaatgcga
agctgtattgcaaccgcggtacggttaattccaagcttaggaaactagatgatgcaatagaagactgcacaaatgcagtgaagcttgat
gacacttacataaaagcctacttgagaagagctcagtgttacatggacacagaacagtatgaagaagcagtacgagactatgaaaaag
tataccagacagagaaaacaaaagaacacaaacagctcctaaaaaatgcgcagctggaactgaagaagagtaagaggaaagattacta
caagattctaggagtggacaagaatgcctctgaggacgagatcaagaaagcttatcggaaacgggccttgatgcaccatccagatcgg
catagtggagccagtgctgaggttcagaaggaggaggagaagaagttcaaggaagttggagaggcctttactatcctctctgatccca
agaaaaagactcgctatgacagtggacaggacctagatgaggagggcatgaatatgggtgatttgatccaaacaatatcttcaaggc
attctttggcggtcctggcggcttcagctttgaagcatctggtccagggaatttcttttttcaatttggctga.

## Transient overexpression of DnaJC7 constructs in tau biosensor cells

DnaJC7 KO or Non-Targeting control tau biosensor cells were plated at 500K cells/well in a six-well dish. 24 hr later, cells were transiently transfected with 1 µg of plasmid containing the WT, HPQ, or ALS-associated Ruby-tagged sequences of DnaJC7 using 5 µL of Lipofectamine 2000 (Gibco) transfection reagent in OptiMEM (Gibco) to a final volume of 125 µL. The Ruby control cells were transfected

with 500 ng of plasmid expressing only Ruby using 5 µL of Lipofectamine 2000 transfection reagent in OptiMEM to a final volume of 125 µL. The vehicle control cells were treated with 5 µL of Lipofectamine 2000 transfection reagent in 120 µL OptiMEM. 48 hr after transfection, the cells were plated in 96-well plates at 20,000 cells/well in 100 µL of media for a standard naked seeding assay. Wells were run on the flow cytometer to completion to ensure collection of a sufficient number of cells with Ruby signal. One-way analysis of variance with Dunnett's correction for multiple comparisons was used to assess statistical significance relative to Ruby-alone controls for overexpression and rescue experiments and relative to the WT C7 construct for rescue with ALS-associated mutants in GraphPad Prism.

## Immunoblotting of DnaJC7 and DnaJB6 from cell lysates

HEK293T tauRD biosensor cell pellets were lysed by resuspension in cold 0.01% Triton in PBS containing cOmplete mini EDTA-free protease inhibitor tablet (Roche) at a concentration of 10% wt/vol followed by incubation in the lysis buffer for 15 min on ice. Homogenates were then clarified by centrifugation at 4°C at a speed of 17,200 RCF for 15 min. The supernatant was then isolated, and total cell lysate protein concentrations were measured using the Pierce BCA Protein Assay Kit (Thermo Fisher).

Ten µg of total cell lysate protein was prepared in 1× (final) LDS Bolt buffer (Invitrogen) supplemented with 10% b-mercaptoethanol and heated for 5 min at 98°C. The proteins were resolved by SDS-PAGE using Novex NuPAGE pre-cast gradient Bis-Tris acrylamide gels (4–12%) (Invitrogen). After gel electrophoresis, resolved proteins were transferred onto Immobilon-P PVDF membranes (MilliporeSigma) using a Bio-Rad Trans-blot semi-dry transfer cell. The membranes were then blocked in 1× TBST buffer (10 mM Tris, 150 mM NaCl, pH 7.4, 0.05% Tween-20) containing 5% non-fat milk powder (Bio-Rad). Membranes were then probed with antibody in TBST containing 5% milk powder.

The following antibodies were used for immunoblotting: rabbit polyclonal anti-DnaJC7 (Proteintech, 11090-1-AP) at a 1:2000 dilution; rabbit polyclonal anti-DnaJB6 (Proteintech, 11707-1-AP) at a 1:2000 dilution; rabbit polyclonal anti-Beta-Tubulin (Proteintech, 10094-1-AP) at a 1:5000 dilution; a secondary donkey-anti-rabbit HRP-linked F(ab')2 (Cytiva, NA9340-1ML) at a 1:5000 dilution when blotting for DnaJC7, a 1:4000 dilution when blotting for DnaJB6, and a 1:5000 dilution when blotting for Tubulin; mouse monoclonal anti-Beta-Actin (Proteintech, 66009-1-Ig) at a 1:5000 dilution; mouse monoclonal anti-GAPDH (Proteintech, 60004-1-Ig) at a 1:10,000 dilution; and a secondary goat-anti-mouse H&L (HRP) (Abcam, ab6789) at a 1:5000 dilution when blotting for Beta-Actin and 1:10,000 when blotting for Beta-Tubulin.

Normalized band intensities for the ALS-associated mutant constructs were calculated using Fiji image analysis software (*Schindelin et al., 2012*). Background signal was subtracted from each DnaJC7 and Beta-Actin control band. The DnaJC7 signal for each construct was then normalized to its corresponding Beta-Actin control band. Finally, the signals for all constructs were normalized to the signal of the WT DnaJC7 construct.

## Calculation of ddG for ALS mutants

Our structural model of DnaJC7 was built in ROSETTA using homology modeling using DnaJC3 (PDB ID: 3IEG) as a template and minimized using the relax protocol (*Song et al., 2013*; *Simons et al., 1999*). A low scoring model was used to evaluate the energetics of ALS-associated mutations in DnaJC7. The 17 ALS-associated mutations were then individually introduced into the initial DnaJC7 model. For each mutation, 35 replicate simulations were run in parallel for WT and mutant DnaJC7 until convergence. To estimate the ddG of monomer, the mean free energy difference between the WT and mutant DnaJC7 structures was calculated (*Barlow et al., 2018*; *Wydorski et al., 2021*). Simulations were performed on the BioHPC computing cluster at UT Southwestern Medical Center. The relax protocol and Flex ddG used Rosetta v3.13 and v3.12, respectively.

## Acknowledgements

Work in the LAJ lab was supported by an NIH-NIA grant RF1AG078888. Work in the MID lab was supported by the following grants: NIH-NIA 3R01AG048678, NIH-NIA 1RF1AG059689, and NIH-NIA 1RF1AG065407. The CLW lab was supported by the McCune Foundation and the Winspear Family Center for Research on the Neuropathology of Alzheimer's Disease. LAJ, CLW, and MID were supported by the Chan Zuckerberg Initiative 2018-191983 and Chan Zuckerberg Initiative

2021-237348. We would like to thank the UT Southwestern Alzheimer's Disease Center for providing pathological brain tissue samples. We also thank the Proteomics Core Facility and Moody Foundation Flow Cytometry Facility at the University of Texas Southwestern Medical Center.

## Additional information

### Funding

| Funder | Grant reference number | Author |
|---|---|---|
| National Institute on Aging | RF1AG078888 | Valerie Ann Perez<br>Vishruth Mullapudi<br>Lukasz A Joachimiak |
| National Institute on Aging | 3R01AG048678 | Valerie Ann Perez<br>David W Sanders<br>Ayde Mendoza-Oliva<br>Barbara Elena Stopschinski<br>Marc I Diamond |
| National Institute on Aging | 1RF1AG059689 | Valerie Ann Perez<br>David W Sanders<br>Ayde Mendoza-Oliva<br>Barbara Elena Stopschinski<br>Charles L White<br>Marc I Diamond |
| National Institute on Aging | 1RF1AG065407 | Valerie Ann Perez<br>Marc I Diamond<br>Charles L White<br>Lukasz A Joachimiak |
| McCune Foundation | | Charles L White |
| Winspear Family Center for Research on the Neuropathology of Alzheimer's Disease | | Charles L White |
| Chan Zuckerberg Initiative | 2018-191983 | Charles L White<br>Lukasz A Joachimiak<br>Marc I Diamond |
| Chan Zuckerberg Initiative | 2021-237348 | Charles L White<br>Marc I Diamond<br>Lukasz A Joachimiak |

The funders had no role in study design, data collection and interpretation, or the decision to submit the work for publication.

### Author contributions

Valerie Ann Perez, Conceptualization, Resources, Data curation, Formal analysis, Investigation, Visualization, Methodology, Writing - original draft, Writing - review and editing; David W Sanders, Conceptualization, Resources, Data curation, Formal analysis, Validation, Investigation, Visualization, Methodology, Writing - original draft, Writing - review and editing; Ayde Mendoza-Oliva, Resources, Validation; Barbara Elena Stopschinski, Conceptualization, Resources, Supervision; Vishruth Mullapudi, Software, Methodology; Charles L White, Resources; Lukasz A Joachimiak, Marc I Diamond, Conceptualization, Supervision, Funding acquisition, Writing - original draft, Project administration, Writing - review and editing

### Author ORCIDs

Valerie Ann Perez  http://orcid.org/0000-0001-8854-515X
David W Sanders  http://orcid.org/0000-0002-1835-6895
Ayde Mendoza-Oliva  https://orcid.org/0000-0002-8917-0940
Barbara Elena Stopschinski  http://orcid.org/0000-0002-5715-4567
Charles L White  http://orcid.org/0000-0002-3870-2804
Lukasz A Joachimiak  http://orcid.org/0000-0003-3061-5850

Marc I Diamond [ORCID] http://orcid.org/0000-0002-8085-7770

**Decision letter and Author response**
Decision letter https://doi.org/10.7554/eLife.86936.sa1
Author response https://doi.org/10.7554/eLife.86936.sa2

## Additional files

### Supplementary files
• MDAR checklist

### Data availability
All data generated or analyzed during this study are included in the manuscript and supporting file. Source data files for included western blot and SDS-PAGE gel images are provided as *Figure 1—figure supplement 1—source data 1*, *Figure 2—figure supplement 1—source data 1*, *Figure 3—figure supplement 1—source data 1*, and *Figure 5—figure supplement 1—source data 1*. Source data files have been provided for Figures 1 (Source Data 1 and 2) and 2 (Source Data 3) on Dryad at: https://doi.org/10.5061/dryad.fj6q57402 and FCS files are deposited on FlowRepository. Experiments for this project are tagged 'Perez_et_al_eLife_2023' and can be retrieved collectively (while logged in with a free account) at http://flowrepository.org/experiments?label=Perez_et_al_eLife_2023 if required.

The following datasets were generated:

| Author(s) | Year | Dataset title | Dataset URL | Database and Identifier |
|---|---|---|---|---|
| Perez VA, Sanders DW, Mendoza-Oliva A, Stopschinski BE, Mullapudi V, White III CL, Joachimiak LA, Diamond MI | 2023 | DnaJC7 specifically regulates tau seeding | https://dx.doi.org/10.5061/dryad.fj6q57402 | Dryad Digital Repository, 10.5061/dryad.fj6q57402 |
| Perez VA, Sanders DW, Mendoza-Oliva A, Stopschinski BE, Mullapudi V, White CL, Joachimiak LA, Diamond MI | 2023 | Figure_3_C | http://flowrepository.org/id/FR-FCM-Z6LD | FlowRepository, FR-FCM-Z6LD |
| Perez VA, Sanders DW, Mendoza-Oliva A, Stopschinski BE, Mullapudi V, White CL, Joachimiak LA, Diamond MI | 2023 | Figure_3_S1_Group_1-A | http://flowrepository.org/id/FR-FCM-Z6NQ | FlowRepository, FR-FCM-Z6NQ |
| Perez VA, Sanders DW, Mendoza-Oliva A, Stopschinski BE, Mullapudi V, White CL, Joachimiak LA, Diamond MI | 2023 | Figure_3_S1_Group_1-B | http://flowrepository.org/id/FR-FCM-Z6NR | FlowRepository, FR-FCM-Z6NR |
| Perez VA, Sanders DW, Mendoza-Oliva A, Stopschinski BE, Mullapudi V, White CL, Joachimiak LA, Diamond MI | 2023 | Figure_3_S1_Group_2-A | http://flowrepository.org/id/FR-FCM-Z6NS | FlowRepository, FR-FCM-Z6NS |
| Perez VA, Sanders DW, Mendoza-Oliva A, Stopschinski BE, Mullapudi V, White CL, Joachimiak LA, Diamond MI | 2023 | Figure_3_S1_Group_2-B | http://flowrepository.org/id/FR-FCM-Z6NU | FlowRepository, FR-FCM-Z6NU |

*Continued on next page*

*Continued*

| Author(s) | Year | Dataset title | Dataset URL | Database and Identifier |
|---|---|---|---|---|
| Perez VA, Sanders DW, Mendoza-Oliva A, Stopschinski BE, Mullapudi V, White CL, Joachimiak LA, Diamond MI | 2023 | Figure_3_S1_Group_3-A | http://flowrepository.org/id/FR-FCM-Z6NV | FlowRepository, FR-FCM-Z6NV |
| Perez VA, Sanders DW, Mendoza-Oliva A, Stopschinski BE, Mullapudi V, White CL, Joachimiak LA, Diamond MI | 2023 | Figure_3_S1_Group_3-B | http://flowrepository.org/id/FR-FCM-Z6XZ | FlowRepository, FR-FCM-Z6XZ |
| Perez VA, Sanders DW, Mendoza-Oliva A, Stopschinski BE, Mullapudi V, White CL, Joachimiak LA, Diamond MI | 2023 | Figure_3_S1_Group_4-A | http://flowrepository.org/id/FR-FCM-Z6XY | FlowRepository, FR-FCM-Z6XY |
| Perez VA, Sanders DW, Mendoza-Oliva A, Stopschinski BE, Mullapudi V, White CL, Joachimiak LA, Diamond MI | 2023 | Figure_3_S1_Group_4-B | http://flowrepository.org/id/FR-FCM-Z6X2 | FlowRepository, FR-FCM-Z6X2 |
| Perez VA, Sanders DW, Mendoza-Oliva A, Stopschinski BE, Mullapudi V, White CL, Joachimiak LA, Diamond MI | 2023 | Figure_3_S1_Group_5-A | http://flowrepository.org/id/FR-FCM-Z6X3 | FlowRepository, FR-FCM-Z6X3 |
| Perez VA, Sanders DW, Mendoza-Oliva A, Stopschinski BE, Mullapudi V, White CL, Joachimiak LA, Diamond MI | 2023 | Figure_3_S1_Group_5-B | http://flowrepository.org/id/FR-FCM-Z6X4 | FlowRepository, FR-FCM-Z6X4 |
| Perez VA, Sanders DW, Mendoza-Oliva A, Stopschinski BE, Mullapudi V, White CL, Joachimiak LA, Diamond MI | 2023 | Figure_3_S1_Group_6-A | http://flowrepository.org/id/FR-FCM-Z6X5 | FlowRepository, FR-FCM-Z6X5 |
| Perez VA, Sanders DW, Mendoza-Oliva A, Stopschinski BE, Mullapudi V, White CL, Joachimiak LA, Diamond MI | 2023 | Figure_3_S1_Group_6-B | http://flowrepository.org/id/FR-FCM-Z6X6 | FlowRepository, FR-FCM-Z6X6 |
| Perez VA, Sanders DW, Mendoza-Oliva A, Stopschinski BE, Mullapudi V, White CL, Joachimiak LA, Diamond MI | 2023 | Figure_4 | http://flowrepository.org/id/FR-FCM-Z6NM | FlowRepository, FR-FCM-Z6NM |
| Perez VA, Sanders DW, Mendoza-Oliva A, Stopschinski BE, Mullapudi V, White CL, Joachimiak LA, Diamond MI | 2023 | Figure_5_A_B | http://flowrepository.org/id/FR-FCM-Z6NX | FlowRepository, FR-FCM-Z6NX |

*Continued on next page*

*Continued*

| Author(s) | Year | Dataset title | Dataset URL | Database and Identifier |
|---|---|---|---|---|
| Perez VA, Sanders DW, Mendoza-Oliva A, Stopschinski BE, Mullapudi V, White CL, Joachimiak LA, Diamond MI | 2023 | Figure_5_D | http://flowrepository.org/id/FR-FCM-Z6NP | FlowRepository, FR-FCM-Z6NP |

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
