## [Editor Report]

This important piece of work identified DNAJC7 as a key J-protein that suppresses seeded tau aggregation. The evidence supporting this work is solid. The work will be of interest to cell biologists and neuroscientists.

---

## [Decision Letter]

**Decision letter after peer review:**

Thank you for submitting your article "DnaJC7 specifically regulates tau seeding" for consideration by *eLife*. Your article has been reviewed by 2 peer reviewers, and the evaluation has been overseen by a Reviewing Editor and Volker Dötsch as the Senior Editor.

Essential revisions:

1) In Supplemental Figure 1 the authors demonstrate two successful DNAJC7 knockout lines. In Figure 2D it would be useful to see a similar phenotype from this independent KO line.

2) Some validation of knockout of the JDPs in Figure 3 should be performed, similar to Supplemental Figure 1.

Related to Figure 3 The statement "DnaJC7 KO enabled more intracellular aggregation than DanJB6 KO" seems too strong considering that statistically significant differences are observed only for a fragment of the plot for 24 hours of the seeding assay

3) In Figure 4 deletion of DNAJC7 and DNAJB6 has differential effects in naked seeding versus Lipofectamine seeding for DS9 and DS10 cell lysates. This should be discussed.

Related to Figure 4A: The effectiveness of AD seeding for C7 KO is very low, considering the results obtained for the NT control. How reliable is < 1 % fraction of FRET positive 'naked' cells treated with AD?

The interpretation of this result in the text does not reflect this ambiguity. Furthermore, the text description of how experiments presented in Figure 4 were performed and interpreted needs improvement- more information.

4) In Figure 4 the authors conclude that DNAJC7 KO increases seeding from tauopathy brains, however, the results only demonstrate an effect in AD brains as PSP and CBD brain lysates fail to induce signal. The text should be modified to reflect this.

Related to Figure 4: text description of how experiments presented in Figure 4 were performed and interpreted needs improvement- more information. Also how reliable is less than 1 % fraction of 'naked' cells treated.

5) When discussing Figure 5A, B it is worth noting that there is evidence that chaperones can bind to the TPR domain of DNAJC7 as well. This is interesting in light of the dominant negative effect the authors describe.

6) In Figure 5D the control of the knockout cell line alone should be included.

Related to Figure 5D: The Seed Concentration scale is in M, while the legend to panel E indicates 33 nM concentration of tau fibrils. By changing the scale on D to nM it will be easier to compare the results from panels D and E.

7) In Supplement Figure 5C the levels of DNAJC7 should be quantified and discussed. The authors do a nice job of utilizing Rosetta to predict changes in stability, however, the data from the western blot don't completely correlate with these predictions (as expected).

8) Supplemental Figure 5 is called Supplemental Figure 4 in the manuscript. This figure is also largely described in the discussion instead of in the Results section. All data should be described in the Results section.

9) Introduction: "We have now tested…and the role of Hsp70 in the regulation of intracellular tau aggregation"- The statement about the role of Hsp70 is too strong. The role of Hsp70 was not tested; the importance of JDP/Hsp70 functional interaction was suggested/implicated by using HPD mutant of C7.

10) "DnaJB6 and DnaJC7 regulation of tau aggregation"

– The first paragraph in this section is confusing- do the authors agree or disagree with the cited results? I think the confusion will be omitted by changing the order: first results obtained in this manuscript- next the results described in the literature.

Related: – the second paragraph of this section can be also improved – see my above comments on the difference between C7 and C6 KOs in increasing tau aggregation

11) The manuscript is written for specialists in the field of cell biology of molecular chaperones. For example: in the Introduction more information about JDP/Hsp70/substrate interaction is needed- to explain the effect of HPD mutant in the Results.

*Reviewer #1 (Recommendations for the authors):*

In this manuscript, the authors assess the composition of the insoluble fraction of a cellular lysate from cells containing aggregated tau repeat domain. From this analysis, the authors identify DNAJC7 as a protein that restricts tau seeding in dividing cells. To further test the role of J-proteins in tau seeding the authors next performed a targeted CRISPR/Cas9 screen to assess the role of each J-protein in alter tau seeding. This resulted in the additional identification of DNAJB6 as a second J-protein that regulated tau seeding. Finally, the authors went on to assess how pathogenic variants of DNAJC7 alter tau seeding.

Overall, one strength of this study is the approaches utilized to identify modifiers of tau seeding. The use of CRIPSR/Cas9 to generate knockouts of the hits from the proteomic screen and to systematically determine the role of J-proteins in tau seeding is also a strength. Together the assays utilized do a thorough job of analyzing the role of DNAJC7 in suppressing tau seeding in their biosensor cell line. One area that remains unclear is if neuronal cells utilize the same array of J-proteins to regulate tau seeding.

One additional strength of the manuscript is the expansion of the studies to assess the seeding capacity of AD, PSP, and CBD patient brain lysates in DNAJC7 and DNAJB6 knockout cells. These results demonstrate that both recombinant tau and AD brain lysates effectively seed tau aggregation in a manner that is suppressed by DNAJC7 and DNAJB6, however, it is unclear how deletion of DNAJC7 and DNAJB6 have differential effects in the naked versus lipofectamine seeding assays with DS9 and DS10 cell lysates. Finally, the authors do a thorough job assessing how pathogenic mutations in DNAJC7 alter its ability to suppress tau seeding. One area of concern is that some mutants that have significant defects in suppressing tau seeding also appear to be expressed at lower steady-state levels.

Specific Comments:

1) In Supplemental Figure 1 the authors demonstrate two successful DNAJC7 knockout lines. In Figure 2D it would be useful to see a similar phenotype from this independent KO line.

2) Some validation of knockout of the JDPs in Figure 3 should be performed, similar to Supplemental Figure 1.

3) In Figure 4 deletion of DNAJC7 and DNAJB6 has differential effects in naked seeding versus Lipofectamine seeding for DS9 and DS10 cell lysates. This should be discussed.

4) In Figure 4 the authors conclude that DNAJC7 KO increases seeding from tauopathy brains, however, the results only demonstrate an effect in AD brains as PSP and CBD brain lysates fail to induce signal. The text should be modified to reflect this.

5) When discussing Figure 5A, B it is worth noting that there is evidence that chaperones can bind to the TPR domain of DNAJC7 as well. This is interesting in light of the dominant negative effect the authors describe.

6) In Figure 5D the control of the knockout cell line alone should be included.

7) In Supplement Figure 5C the levels of DNAJC7 should be quantified and discussed. The authors do a nice job of utilizing Rosetta to predict changes in stability, however the data from the western blot don't completely correlate with these predictions (as expected).

8) Supplemental Figure 5 is called Supplemental Figure 4 in the manuscript. This figure is also largely described in the discussion instead of in the Results section. All data should be described in the Results section.

*Reviewer #2 (Recommendations for the authors):*

J-domain protein (JDP) co-chaperones determine substrate specificity of their Hsp70 partners via direct delivery of specific substrate(s) or by attracting Hsp70 to specific sites of action, under both physiological conditions and stress or pathology conditions. JDP/Hsp70 systems are implicated in many diseases, including neurodegenerative tauopatheis caused by the accumulation of tau protein aggregates toxic to neural cells. Evidence indicates that JDP/Hsp70 systems prevent tau aggregate formation both in vitro and in vivo. However, considering that in the cytosol of eukaryotic cells there are multiple, structurally and functionally divergent, JDPs that partner with a limited number of Hsp70s, it is unclear which JDP(s) are critical for preventing tau aggregation. The authors addressed this question by determining the tau aggregate interactors using a proteomic approach and by systematically testing how knock out (KO) of those interactors affects the formation of tau aggregates in cells. Among 24 interactors tested, only KO of the JDP termed DNAJC7 (C7) resulted in tau aggregate accumulation. Next, the authors KO all 50 JDPs present in humans. Only DNAJB6 (B6), which was previously shown to affect tau aggregation, had an effect like C7. However, only in the C7 KO cells did tau aggregates accumulate upon treatment with brain lysates from patients with Alzheimer's Disease (AD) – although the results obtained for not transfected 'naked' cells are not very convincing. Using a substitution variant (D/Q) in the HPD motif of the J-domain, the authors demonstrated that disruption of the functional interaction between C7 and its Hsp70 partner(s) enhances tau aggregation. Furthermore, the enhancement of tau aggregation is observed regardless of whether a WT copy of C7 was present or absent in the cell – indicating a dominant negative effect of C7(D/Q). Finally, the authors demonstrated that among 17 disease-associated mutants of C7, 5 displayed enhanced tau aggregation, and one of them (T341) displayed a gain of function phenotype similar to the D/Q J-domain variant.

The strengths of this manuscript are (i) the unbiased identification of DnaJC7 as a tau aggregate interactor, which modulates the formation of tau aggregates, (ii) the demonstration that among 50 human JDPs only C7 and C6 prevent the formation of tau aggregates (iii) demonstration that C7/Hsp70 interaction is critical for this function. The authors established the DNAJC7 JDP as an important modulator of tau aggregation in vivo.

The weakness of this manuscript is that the experiments were performed using HEK293 cells not with neural cells – more relevant to the tau pathology. Furthermore, in some cases the authors "overinterpret" the obtained results exaggerating the importance of C7 in comparison to C6 (e.g. description of the results from Figure 3 and Figure 4A). Finally, the writing could be improved by making this manuscript more accessible to readers from outside the molecular chaperones cell biology field and by improving how authors results are discussed in the context of published data on the role of JDP/Hsp70 systems in preventing tau aggregation.

Figure 3 The statement "DnaJC7 KO enabled more intracellular aggregation than DanJB6 KO" seems too strong considering that statistically significant differences are observed only for a fragment of the plot for 24 hours of the seeding assay.

Figure 4A The effectiveness of AD seeding for C7 KO is very low, considering the results obtained for the NT control. How reliable is < 1 % fraction of FRET positive 'naked' cells treated with AD?

The interpretation of this result in the text does not reflect this ambiguity. Furthermore, the text description of how experiments presented in Figure 4 were performed and interpreted needs improvement – more information.

Figure 5D The Seed Concentration scale is in M, while the legend to panel E indicates 33 nM concentration of tau fibrils. By changing the scale on D to nM it will be easier to compare the results from panels D and E.

Introduction: "We have now tested…and the role of Hsp70 in the regulation of intracellular tau aggregation". The statement about the role of Hsp70 is too strong. The role of Hsp70 was not tested; the importance of JDP/Hsp70 functional interaction was suggested/implicated by using HPD mutant of C7.

Results- DS9 cell line is introduced in the legend of Figure 1- Supplement 1 but is not mentioned in the text.

Figure 4 – text description of how experiments presented in Figure 4 were performed and interpreted needs improvement- more information. Also how reliable is less than 1 % fraction of 'naked' cells treated.

Discussion: "The large family of JDPs are thought to be functionally redundant" – a reference is needed.

"DnaJB6 and DnaJC7 regulation of tau aggregation"

– The first paragraph in this section is confusing- do the authors agree or disagree with the cited results? I think the confusion will be omitted by changing the order: first results obtained in this manuscript- next the results described in the literature.

– The second paragraph of this section can be also improved – see my above comments on the difference between C7 and C6 KOs in increasing tau aggregation.

– DS9 cells were introduced for the first time in the legend to Figure 1- Supplement 1 and not mentioned in the Results(?)

Overall writing issues – to me, this manuscript is written for specialists in the field of cell biology of molecular chaperones. For example: in the Introduction more information about JDP/Hsp70/substrate interaction is needed- to explain the effect of HPD mutant in the Results.

---

## [Author Response]

Essential revisions:1) In Supplemental Figure 1 the authors demonstrate two successful DNAJC7 knockout lines. In Figure 2D it would be useful to see a similar phenotype from this independent KO line.

There may be confusion regarding the numbering of the supplementary figures. Figure 1 – Supplement 1 shows the SDS-PAGE gels for the partial purification of tau aggregates. Figure 2 – Supplement 1 shows the requested Western Blot analysis confirming the KO of DnaJC7 for the cell line used in Figure 2D.

2) Some validation of knockout of the JDPs in Figure 3 should be performed, similar to Supplemental Figure 1.

We have added an additional panel to Figure 3 – Supplement 1 (Panel C) showing Western Blot analysis confirming the partial and full knockouts of the top hits from the JDP CRISPR screen, DnaJC7 and DnaJB6. Validation of all other JDP knockouts was validated only by survival in puromycin-containing media. The DnaJC7 KO cell line was subsequently remade and the full KO cell line is shown in Figure 5 – Supplement 1 Panel B.

Related to Figure 3 The statement "DnaJC7 KO enabled more intracellular aggregation than DanJB6 KO" seems too strong considering that statistically significant differences are observed only for a fragment of the plot for 24 hours of the seeding assay

We have changed the phrasing in this section to reflect that the differences in seeding by the DnaJB6 and DnaJC7 are only significant at two tau concentrations.

3) In Figure 4 deletion of DNAJC7 and DNAJB6 has differential effects in naked seeding versus Lipofectamine seeding for DS9 and DS10 cell lysates. This should be discussed.Related to Figure 4A: The effectiveness of AD seeding for C7 KO is very low, considering the results obtained for the NT control. How reliable is < 1 % fraction of FRET positive 'naked' cells treated with AD?The interpretation of this result in the text does not reflect this ambiguity. Furthermore, the text description of how experiments presented in Figure 4 were performed and interpreted needs improvement- more information.

Additional discussion on the differential effects of DnaJC7 vs. DnaJB6 KO on DS9 and DS10 cell lysate seeding has been added in both the Results and Discussion sections of the text. A full description of the experimental setup for Figure 4 can be found in the Methods section. Additional details of the experiment were omitted from the main text for clarity and brevity.

Naked seeding results in markedly reduced seeding efficiency compared to Lipofectamine-mediated seeding. However, in our experience, we do not see false-positive seeding when performing naked seeding. Therefore, we believe that the < 1% seeding reported for the DnaJC7 KO cell line seeded with AD brain lysates is a true positive result that is well above the noise threshold for naked seeding with brain lysates.

4) In Figure 4 the authors conclude that DNAJC7 KO increases seeding from tauopathy brains, however, the results only demonstrate an effect in AD brains as PSP and CBD brain lysates fail to induce signal. The text should be modified to reflect this.Related to Figure 4: text description of how experiments presented in Figure 4 were performed and interpreted needs improvement- more information. Also how reliable is less than 1 % fraction of 'naked' cells treated.

Upon closer inspection, we realized we had incorrectly used two-way ANOVA for the statistical analysis of several of our experiments. The analysis has now been corrected to use one-way ANOVA with Dunnett’s correction for multiple comparisons and now highlights that the DnaJC7 KO did significantly increase tau seeding for all tauopathy brains tested.

Please see our response to Comment #3 above regarding the interpretation of naked seeding results.

5) When discussing Figure 5A, B it is worth noting that there is evidence that chaperones can bind to the TPR domain of DNAJC7 as well. This is interesting in light of the dominant negative effect the authors describe.

The different Hsp70 and Hsp90 binding sites found on the TPR domains of DnaJC7 are mentioned in the text. Additional discussion on the potential roles of mutations to these binding sites is provided in the Discussion section.

6) In Figure 5D the control of the knockout cell line alone should be included.Related to Figure 5D: The Seed Concentration scale is in M, while the legend to panel E indicates 33 nM concentration of tau fibrils. By changing the scale on D to nM it will be easier to compare the results from panels D and E.

The Vehicle Control represents the control of the parent DnaJC7 knockout cell line, shown in Figure 5 – Supplement 1D. This and other controls were omitted from the main figure for clarity.

Additionally, we have changed the scale of the x-axis in all panels of Figure 5 and its supplement to be in nM for easier comparison to panel E.

7) In Supplement Figure 5C the levels of DNAJC7 should be quantified and discussed. The authors do a nice job of utilizing Rosetta to predict changes in stability, however, the data from the western blot don't completely correlate with these predictions (as expected).

We have quantified the band intensities for the ALS-mutant constructs in Supplemental Figure 5C and added this quantification as Supplemental Figure 5F and 5G.

8) Supplemental Figure 5 is called Supplemental Figure 4 in the manuscript. This figure is also largely described in the discussion instead of in the Results section. All data should be described in the Results section.

We have changed the labeling of all supplemental figures to a consistent naming scheme and have fixed any instances of Supplemental Figure 5 (now Figure 5 – Supplement 1) misnumbering.

Additionally, we have now introduced the results presented in Supplementary Figure 5 in the main Results text.

9) Introduction: "We have now tested…and the role of Hsp70 in the regulation of intracellular tau aggregation"- The statement about the role of Hsp70 is too strong. The role of Hsp70 was not tested; the importance of JDP/Hsp70 functional interaction was suggested/implicated by using HPD mutant of C7.

We have changed the wording in this paragraph to reflect this discrepancy.

10) "DnaJB6 and DnaJC7 regulation of tau aggregation"– The first paragraph in this section is confusing- do the authors agree or disagree with the cited results? I think the confusion will be omitted by changing the order: first results obtained in this manuscript- next the results described in the literature.Related: – the second paragraph of this section can be also improved – see my above comments on the difference between C7 and C6 KOs in increasing tau aggregation

We have reordered the paragraphs in the "DnaJB6 and DnaJC7 regulation of tau aggregation" section of the Discussion and added additional discussion on the differences in DnaJC7 vs. DnaJB6 KO in affecting tau seeding from different sources.

11) The manuscript is written for specialists in the field of cell biology of molecular chaperones. For example: in the Introduction more information about JDP/Hsp70/substrate interaction is needed- to explain the effect of HPD mutant in the Results.

Additional description of the JDP/Hsp70/substrate interaction and the role of the HPD motif has been added to the Introduction.